# High Structural Diversity of Aeruginosins in Bloom-Forming Cyanobacteria of the Genus *Planktothrix* as a Consequence of Multiple Recombination Events

**DOI:** 10.3390/md21120638

**Published:** 2023-12-13

**Authors:** Elisabeth Entfellner, Kathrin B. L. Baumann, Christine Edwards, Rainer Kurmayer

**Affiliations:** 1Research Department for Limnology, Universität Innsbruck, Mondseestrasse 9, 5310 Mondsee, Austria; elisabeth.entfellner@gmx.at (E.E.); katbaumann87@gmail.com (K.B.L.B.); 2CyanoSol Research Group, Pharmacy & Life Sciences, Robert Gordon University, Aberdeen AB10 7GJ, UK; c.edwards@rgu.ac.uk

**Keywords:** horizontal gene transfer, chemical diversification, freshwater ecotypes, shallow waterbodies, deep waterbodies

## Abstract

Many compounds produced by cyanobacteria act as serine protease inhibitors, such as the tetrapeptides aeruginosins (Aer), which are found widely distributed. The structural diversity of Aer is intriguingly high. However, the genetic basis of this remains elusive. In this study, we explored the genetic basis of Aer synthesis among the filamentous cyanobacteria *Planktothrix* spp. In total, 124 strains, isolated from diverse freshwater waterbodies, have been compared regarding variability within Aer biosynthesis genes and the consequences for structural diversity. The high structural variability could be explained by various recombination processes affecting Aer synthesis, above all, the acquisition of accessory enzymes involved in post synthesis modification of the Aer peptide (e.g., halogenases, glycosyltransferases, sulfotransferases) as well as a large-range recombination of Aer biosynthesis genes, probably transferred from the bloom-forming cyanobacterium *Microcystis*. The Aer structural composition differed between evolutionary *Planktothrix* lineages, adapted to either shallow or deep waterbodies of the temperate climatic zone. Thus, for the first time among bloom-forming cyanobacteria, chemical diversification of a peptide family related to eco-evolutionary diversification has been described. It is concluded that various Aer peptides resulting from the recombination event act in chemical defense, possibly as a replacement for microcystins.

## 1. Introduction

The aeruginosins (Aer) constitute a bioactive peptide family that is found widely distributed among marine and freshwater cyanobacteria [1], and also among higher organisms such as marine sponges of the family Dysideidae [2,3]. The structural diversity of Aer is intriguingly high, and structure–function relationships have been used to understand the consequences for inhibition of serine proteases and other enzymes [4,5]. The bloom-forming cyanobacterium *Planktothrix* is a prolific producer of Aer, as well as other peptide families such as the toxic microcystins and bioactive anabaenopetins, cyanopeptolins, microginins, microviridins and prenylagaramides [6]. In particular, the Aer peptide family has the highest structural variability [7,8]. Other Aer peptide producers include the genera *Microcystis*, *Nostoc*, *Nodularia* [3,9] and *Spaerocarvum* [10]. Aer synthesis is encoded by *aer* genes, organized into a biosynthetic gene cluster (*aer* BGC) encoding several non-ribosomal peptide synthetases (NRPS), which are responsible for specific amino acid recognition and incorporation. The resulting Aer molecule is a linear tetrapeptide, containing an unusual 2-carboxy-6-hydroxyoctahydroindole (Choi) moiety. The characteristic amino acid Choi is synthesized via the enzymes *AerD*, *E*, *F* [11,12], modifying an amino acid that is activated by the first adenylation domain of the NRPS *AerG* [13]. In contrast to other peptides (i.e., microcystins, anabaenopeptins), numerous tailoring enzymes (e.g., sulfatases, halogenases, glycosyltransferases, methyltransferases, acetyltransferases) introduce high structural variability among the Aer peptide family [13,14].

Genome sequencing and metabolomics have shown that the Aer peptide family is more widely distributed than previously assumed [14,15]. Besides the structurally related spumigins and pseudoaeruginosins, also dysinosins [16], varlaxins produced by *Nostoc sp.* [17] and suomilide produced by *Nodularia sphaerocarpa* [14] might be considered as members of the Aer peptide family. In particular, the *spuA-F* genes encoding spumigin synthesis show a partial similarity to the *aer* gene cluster [18], while the synthesis of pseudoaeruginosins is encoded through a combination of *spu* and *aer* genes, as described from *Nodularia spumigena* [19]. Suomilide is characterized by the structurally related azabicyclononane (Abn) moiety [20], which is considered as a modification of Choi encoded by *suoK* and *suoH* [14].

The Aer synthesis pathway was first described from *Planktothrix* strain NIVA-CYA126/8 [13], which was found partially divergent from *Microcystis* [21]. *Planktothrix* NIES-204 and *Microcystis* NIES-843 demonstrate high homogeneity of *aer* genes [22]. By comparing the *aerB*, *aerG*, *aerD*, *aerE* and *aerF* genes, sequenced from 103 strains of cyanobacteria (representing twelve genera), a higher similarity of *aer* genes within *Planktothrix, Microcystis* and *Aphanothece* was observed, while *aer* genes from *Nostoc*, *Nodularia* or *Anabaena* were rather dissimilar [14]. Thus, in contrast to microcystin synthesis, but in accordance with anabaenopeptin synthesis [23], a rather high similarity between *aer* genes of *P. agardhii* and *Microcystis* sp. strains has been observed.

Since the *aer* genes have been detected most frequently among various *Planktothrix* species [6], we were interested in exploring the *aer* gene organization and the structural consequences of recombination events within this genus. It was the aim of this study to describe all the recombination events resulting in Aer structural modification in relation to *Planktothrix* spp. eco-evolutionary divergence. Thus, we quantified the variations within *aer* genes among 124 strains and correlated this genetic variability to the characterized Aer variants. The *Planktothrix* strains were isolated from 40 diverse waterbodies located on three different continents, occurring between the tropical and northern temperate climatic zones (Europe, North America and Africa), which were phylogenetically and ecologically assigned previously. In brief, three major lineages were identified: Lineage 1 and 2 included strains assigned to *P. agardhii/P. rubescens,* thriving either in the plankton of shallow or deeper lakes of the temperate climatic zone, while Lineage 3 consisted of strains representing more distantly related species such as *P. pseudagardhii, P. mougeotii* and *P. tepida*, occurring frequently in (sub)tropical regions [23,24].

## 2. Results and Discussion

### 2.1. Aeruginosin Biosynthesis Genes Show Multiple Recombinations and a Variable Evolutionary Origin

As previous studies reported high similarity of *aer* genes between *Planktothrix* and *Microcystis* [14,22,25], we compared the nucleotide sequence of the *aer* genes between these genera in detail (Figure 1 and Appendix A). Regarding their similarity to *Microcystis*, two major groups of *aer* BGCs were observed in *Planktothrix* strains (Figure 1A). Overall, group 1 (e.g., *Planktothrix* NIES-204) showed a higher similarity to *M. aeruginosa* NIES-2481, i.e., the genetic distance on nucleotide level was <0.25. In contrast, group 2 consistently showed higher genetic distance > 0.2. The core *aer* gene composition and arrangement, including all genes encoding NRPS (i.e., *aerA*, *aerB* and *aerG/G1* + *G2/M*), the genes encoding Choi synthesis (*aerD-F*), as well as the ABC transporter (*aerN*), were conserved. However, *aerG* (in group 2) was replaced by *aerG1 and aerM* (in group1) or *aerG1* and *aerG2* (in *M. aeruginosa* NIES-98). In addition, concerning *aerB,* a high nucleotide variation within *Planktothrix* and *Microcystis* was observed, resulting in distinct phylogenetic clades (Appendix A). Accordingly, four different specificity-conferring codes for the adenylation domain of AerB were identified using the webserver NRPS predictor2 [26]: Leu, Ile, Phe and Tyr (Appendix A). Notably, one (benthic) and distantly related *Planktothrix* strain PCC11201 [27] differed substantially in *aer* gene distance from all other *Planktothrix* strains (Figure 1B). Here, the *aer* genes were arranged similarly to those in *Microcystis* with *aerI* (encoding a putative glycosyltransferase), while *aerL*, *aerK* and ORF2 (open reading frame, encoding a putative acetyltransferase) were lacking. Consequently, in this specific—rather distantly related—*Planktothrix* strain PCC11201, the *aer* BGC might have its own origin from another unknown cyanobacterium.

The accessory genes *aerC*, *aerH-L* and ORF1-9 were found more irregularly distributed. The genes *aerK*, ORF1 (a putative oxidoreductase) and ORF2 (a putative acetyltransferase) were found among strains assigned to group 1 only. Notably, *aerK,* encoding a putative isomerase [14], had high similarity to *Microcystis* (genetic dissimilarity < 0.1), but was present in full length only in *Microcystis* and the *Planktothrix* strains No980 and No1020, whereas most other *Planktothrix* strains of group 1 had remnants of it. ORF1 was present in all strains of group 1 and showed high similarity (distance < 0.15; except for strain PCC11201). ORF2 was found in most strains of group 1; however, it was lacking in *Planktothrix* No1020 and PCC11201, and was found interrupted by a stop codon in *Microcystis* strain NIES-2549. The genes *aerJ* (encoding a halogenase) and *aerL* (encoding a sulfotransferase) occurred with highest frequency among strains carrying *aer* BGC group 1. Phylogenetic analysis of *aerL* sequences (372–1011 bp) revealed four clusters (Appendix A), possibly indicating sulfation at different moieties of the Aer molecule: Sulfation at Hpla (e.g., *Microcystis* NIES-843 [21]), sulfation at Choi (e.g., *Microcystis* NIES-98, [28]) and sulfation at an unknown position (e.g., *Planktothrix* No365). In contrast to group 1, the a*erC*, *H*, *I* and ORF3-9 genes were found frequently among strains carrying *aer* BGC group 2. However, only ORF8 (encoding an Aldo-/Keto-reductase, [14]) occurred consistently among strains that carried *aerG* instead of *aerM/G2* (=strains carrying *aer* BGC group 1) and might be essential for Aer synthesis (Table 1).

Last but not least, *Planktothrix* strains No713 and PCC9214, assigned to Lineage 3, carried an unknown NRPS/PKS gene cluster, comprising *aerD-F* genes possibly encoding Choi synthesis (Figure 1B and Appendix A). These strains may produce unknown peptides carrying a Choi moiety further extending the Aer peptide family.

### 2.2. Aer Gene Organization and Composition in Relation to Planktothrix Evolution

The *aer* gene composition was compared among 124 *Planktothrix* strains, which were phylogenetically assigned to three major lineages [23,24]: Lineage 1 (*n* = 42)/1A (*n* = 10) comprised strains isolated from plankton of shallow lakes mostly from the temperate climatic zone. In contrast, Lineage 2 (*n* = 45)/2A (*n* =16) comprised strains isolated from deeper physically stratified lakes of the temperate climatic zone, while Lineage 3 (*n* = 11) comprised strains isolated from the (sub)tropical climatic zone [23,24]. The presence/absence of *aer* genes was determined via PCR, amplifying 2 kbp-sized fragments of the *aer* gene cluster without interruption (Appendix A). All strains assigned to Lineage 1 or 2 showed at least the PCR products indicative of *aerA* (Appendix A). Out of all 124 strains, the *aer* genes were detected for 115 strains (93%). The strains were grouped into three clusters via principal component analysis (PCA), (Figure 2): Cluster I contained 24 strains (assigned to Lineage 1 and 3), representing the genotypes carrying the *aer* BGC similar to *Microcystis* (i.e., strains carrying *aer* BGC group 1, Figure 1A). Cluster II comprised the majority of strains from all lineages (*n* = 91, i.e., strains carrying *aer* BGC group 2, Figure 1A), wherein a subgroup (*n* = 7) showed large gene cluster deletions. Cluster III contained nine strains from Lineage 3, all lacking the *aer* genes.

### 2.3. Aeruginosin Structural Variation and Chemical Diversification in Relation to Phylogenetic Lineages

We used HPLC-MS^n^ to detect and characterize the Aer peptides, as extracted from biomass of the 124 *Planktothrix* strains [24]). In order to characterize the Aer structures in detail, the amino acid sequence of the tetrapeptide backbone, modifications (e.g., sulfates, chlorines, methyl, acetyl and prenyl moieties) and partly also the position of these additional groups were analyzed by specific MS^n^ fragmentation (Appendix A). A number of Aer peptide variants, sometimes even co-synthesized by the same strain, had identical protonated masses and fragmentation pattern, but eluted at different retention times. After re-isolation from chromatographic separation, the Aer peptide variants still eluted at different times, indicating isomers (e.g., Appendix A strain No778).

An exact identification was possible only for a few known Aer peptides, such as Aer 126A/B from *P. agardhii* NIVA-CYA126/8 [13], Aer 828A from *P. rubescens* No91/1 [30] and Aer 89A/B from *M. aeruginosa* NIES-89 [31], i.e., from strain No1020 (Lineage 3). Some other Aer variants were detected among *Planktothrix* sp. previously, however, without structural elucidation. Furthermore, we detected many unknown structural Aer variants, partly variations of known Aer peptides carrying the same tetrapeptide backbone (Appendix A).

We found 79 *Planktothrix* strains (64% of all strains) as Aer producers (Appendix A). The proportion of Aer producers varied between lineages, i.e., strains assigned to Lineage 1/1A showed the highest proportion of Aer containing strains (48 out of 52 strains), while only 29 out of 61 strains (48%) assigned to Lineage 2/2A and 2 out of 11 strains (18%) assigned to Lineage 3 were found active as Aer producers (Figure 3A,B, Appendix A). In total, 73 Aer variants were differentiated. Alltogether, strains contained between one and nine (median = 3) Aer variants (i.e., strain NIES-596 contained nine structural Aer variants). Strains assigned to Lineage 1 contained up to nine (median = 3) variants, while strains assigned to Lineage 2 contained less Aer structural variants (up to five). The majority of Aer variants (56 variants, 77%) occurred in more than one strain; however, only 27 Aer variants (37%) occurred in more than three strains. The most frequent Aer variants (occurring in 10–15 strains) constituted the putative Aer variants [M+H]^+^ 717.3, 715.3, 771.5 and 805.5 (Appendix A) related to the large-range recombination event (*aerJ*–*aerN*, >20 kbp, *aer* BGC group 1 in Figure 1). Only seven variants were shared between Lineages 1 and 2 (Figure 3C). Typically, strains inactive in Aer synthesis showed either a large-range deletion of *aer* genes or the presence of an insertion sequence (IS) element (Appendix A).

In summary, only strains assigned to Lineage 1 and 3 carried the large-range *aer* gene recombination resulting in the most frequent structural variant Aer 716 (partly co-produced with Aer 688) (Appendix A). Interestingly, many of these strains of Lineage 1 and 3 have lost the genes encoding microcystin synthesis and have become nontoxic, as reported previously [6], e.g., only 11% of the strains of Lineage 1 carry *mcy* genes and produce MC (6 out of 52 strains). In contrast, strains of Lineage 2 were more frequently inactive in Aer production, however, all the strains carry the *mcy* genes and 70% produce MC (18 out of 61 strains are inactive), Appendix A.

### 2.4. Relationships between Aeruginosin Structural Modification and Core/Accessory Aer Genes

Little variation was found in the tetrapeptide (backbone) structure, detecting five structural variants (without differentiating Leu from Ile), i.e., Plac–Leu(Ile)–Choi–Argal, Hpla/Plac–Leu(Ile)–Choi–Agm and Hpla/Plac–Phe–Choi–Agm (Figure 3C). We could not find a relationship between any accessory *aer* gene and the catalysis of the hydroxylation of Plac to Hpla; therefore, a relaxed specificity of the adenylation domain of AerA was assumed. In position 2, we found either Leu (e.g., Aer 126A/B) or Ile (which is indistinguishable from Leu by means of MS^n^ fragmentation) or Phe (e.g., Microcin SF608). Among the Arg derivates at pos. 4, we found either Argal (e.g., Aer 89) or Agm (e.g., Aer 126B), though frequently Agm was found modified to Aeap (e.g., Aer 126A). Most strains showed a specific incorporation of only one Arg derivative, and only four strains produced Aer variants in parallel containing either argininal or Agm (e.g., NIVA-CYA116).

Increased structural diversity was generated by accessory modifications: chlorination at pos. 1 and/or 2, methylation at pos. 1, glycosylation at pos. 3, sulfation at pos. 1 or 3 and the conversion of Agm to Aeap at pos. 4. Moreover, we observed putative *O*-acetylation (plus 42 Da, C_2_H_3_O) of Choi and *N*-prenylation (plus 68 Da, C_5_H_8_) of Agm (e.g., Aer KB676 [32]).

No obvious difference in the amino acid specificity conferring code of the adenylation domain of AerA between strains incorporating either only Plac (30%) or Plac/Hpla (15%) or only Hpla (55%) at pos. 1 (Plac/Hpla) could be identified (i.e., for prediction of amino acid activation, no precedent in databases [26] predicting the substrate of adenylation domains could be found). The observed accessory modifications of Hpla at pos. 1 consisted of chlorination, methylation or sulfation. Chlorination was found catalyzed by the halogenase encoded in *aerJ,* as described for *M. aeruginosa* PCC7806 [21,29]. When compared to the *aer* genes from *Planktothrix* strain NIVA-CYA126/8, an additional ORF9 was located between ORF7 and 8 which was identified as a putative methyltransferase. Indeed, methylated Aer variants were detected in some of the strains carrying this additional ORF9 (Figure 4). Previously, in *M. aeruginosa*, the *aerL* gene was identified as a putative sulfotransferase, whereby in *M. aeruginosa* NIES-843, the hydroxyl group of Hpla at pos. 1 was found sulfated and in *M. aeruginosa* NIES-98, the Choi at pos. 3 was found sulfated [21]. Correspondingly, in this study, the presence of the *aerL* gene correlated with sulfated Aer. The sulfate group was not detected in initial LC-MS^n^ analysis in the positive MS mode. However, nucleotide analysis of *aerL* revealed two *aerL1* and *aerL2* genotypes (Appendix A), most likely resulting in either sulfation of Choi (e.g., strain No365) or sulfation of Hpla (e.g., No66) at their hydroxyl group.

The sequence analysis of *aerB* (Appendix A) resulted in three different specificity-conferring codes of the catalytic domain of the adenylation domains: DAWFLGNV, predicted to activate Leu (e.g., No66, NIVA-CYA126/8); DAFFLGV, related to Ile (e.g., No790); and DAWTIAAV, related to Phe (e.g., No82). Indeed, we could confirm Phe in Aer structure among strains carrying AerB predicted to activate Phe (e.g., No82, PCC7821, NIVA-CYA98 and NIVA-CYA406).

In contrast to pos. 2, Choi was found conserved at pos. 3. Notably, *aerDEF* genes (encoding Choi synthesis) have a high sequence dissimilarity between *aer* BGC groups 1 and 2, suggesting a different origin (Figure 1). Glycosylation at the hydroxyl group of Choi was commonly detected. Previously, the *aerI* gene was described as a glycosyltransferase catalyzing the transfer of the xylose moiety to Choi resulting in aeruginosides [13]. In addition, a sulfate group was found linked to the xylose moiety by the same authors [13]. Correspondingly, many strains carrying a sulfate group shared the ORF7 encoding a putative sulfotransferase (Figure 4), [14].

The phylogenetic comparison of the homologous genes *aerM*, *aerG2* and the second adenylation domain of *aerG* (Appendix A) overall corresponded to the taxonomy of the genus *Planktothrix* or *Microcystis*, with one exception, i.e., the *aerG2* from *M. aeruginosa* NIES-98 was found related to the second adenylation domain of *aerG* from *Planktothrix* but not to *aerM*. As a result, all *Planktothrix* strains carrying *aerM* showed argininal (25%), whereas the other strains carrying *aerG* contained Agm (70%) at pos. 4 of the Aer peptide. Only four strains (NIVA-CYA116, No259, No281, No307) possibly produced Aer variants with both arginine derivatives. Agm was often found modified to Aeap, as already described for Aer 126A [13]. Taking all Aer-producing strains together, the presence of *aerC* correlated perfectly with this modification. Furthermore, we found some structural variants with an additional moiety at Agm (+68 Da), suggesting *N*-prenylation which, however, could not be related to an accessory enzyme encoded within the *aer* BGC.

In a previous study, a reductase encoded within *aerM* has been suggested to catalyze the cleavage of the Aer molecule from NRPS in *M. aeruginosa* NIES-843 and PCC7806 [21]. Such a reductase was missing in all *Planktothrix* strains carrying *aerG* only. Instead, ORF8 (a putative aldo-keto reductase [14]) has been detected among these strains (Figure 1) which might encode an alternative enzyme for cleavage of the Aer molecule.

According to Ahmed et al. [14], the *aerK* gene may encode an isomerase. This could explain why strain No980 (carrying *aerK*) produces Aer peptides with same mass but different retention time when compared with Aer peptides from strains e.g., No253 and NIVA-CYA116 carrying only *aerK* fragments. Analogously, ORF3 may encode an isomerase as well, since strain No108 produced Aer peptides with same mass but different retention time when compared with strains No97 and No91/1, sharing a similar *aer* gene organization but lacking *aerC* and ORF3.

### 2.5. High Resolution Mass Spectrometry (HRMS)

Six putative chlorinated Aer compounds related to the large-range *aer* gene recombination event (*aerJ*–*aerN*, >20 kbp) were isolated and purified from *Planktothrix* strains No66, NIVA-CYA116 and No1020. The HRMS results confirmed the presence of Choi ([M+H]^+^ 140 or 122) and chlorine in all six purified Aer compounds and showed typical Aer fragmentation patterns (Appendix A). For *Planktothrix* strain No66, a plausible sum formula for purified Aer structure matching the obtained fragmentation pattern was calculated (Table 2). Consequently, it was concluded that the structure of Aer [M+H]^+^ 717 was possibly identical to the Aer structural variant already described from *M. aeruginosa* NIES-89, i.e., Aer 89A/B [31].

### 2.6. Toxicity of Aeruginosins Resulting from the Large-Range Recombination Events

In general, Aer peptides have been found bioactive because of proteolytic enzyme inhibition (i.e., serine proteases such as trypsin, thrombin, plasmin) effective in the nanomolar to lower micromolar range (e.g., [2,3,4,5]). We tested the toxicity of the variants Aer 716 ([M+H]^+^ 717.3 from No1020), Aer 688 ([M+H]^+^ 689.3 from NIVA-CYA116) and Aer 828A ([M+H]^+^ 829.3 from No91/1) using a standard toxicity assay (i.e., the anostracan crustacean *Thamnocephalus platyurus*). The observed toxicity was compared with that of [D-Asp^3^, (E)-Dhb^7^] microcystin (MC)-RR and other bioactive peptides including anabaenopeptin B and F (Appendix A). Among the tested peptides, the [D-Asp^3^, (E)-Dhb^7^] MC-RR showed highest toxicity followed by Aer 828A in a low µM range (e.g., [30], this study). The Aer 716 (resulting from the large-range *aer* gene recombination) had lower toxicity, comparable to anabaenopeptin B. Aer 688 had the lowest toxicity, comparable to anabaenopeptin F.

Notably, Aer 828A carries chlorine at Leu (pos. 2) of the molecule, while Aer 716 and Aer 688 carry a chlorine at Hpla (pos. 1) of the molecule. Scherer et al. [33] tested the effect of an additional chlorine and/or sulfate on Aer derivatives using the same acute toxicity *T. platyurus* assay as applied in this study. Their results indicated a comparable range in toxicity when a sulfate (at Xylose) or a chlorine (at pos. 2) or both were attached, while the same backbone structure lacking both moieties (i.e., Aer 126A) had lower toxicity. Overall, it is very likely that the presence of a chlorine or sulfate plays a role in modulating Aer toxicity [33]. However, by the same authors, conformational changes resulting from a chlorine towards both increasing and decreasing the bioactivity/toxicity also have been suggested. Analogously, since Aer 716 and Aer 688 only differ in Argal vs. Agm (pos. 4), steric effects in modulating the fit of the guanidinium moiety into the catalytic site of proteases might explain the reduced toxicity of Aer 688 [33].

## 3. Conclusions

By comparing the *aer* gene composition among *Planktothrix,* it was possible to show that strains (genotypes) carrying the large-range recombination of *aer* genes formed a more closely related *aer* genotype group that was found distinct from other strains. Very likely, this recombination stemmed from horizontal gene transfer (HGT) and the recombination was found to be integrated in frame and functional. Since the *aerA* and *aerN* genes located at the 5′ end or 3′ end of the *aer* BGC still showed high similarity with *Planktothrix*, but the other *aer* genes were found most similar to *aer* genes from *Microcystis,* an HGT event occurring between the two different genera is plausible.

The elucidation of the genetic basis of Aer synthesis improved our understanding of the evolution of the impressive Aer structural diversity: There was a perfect correlation between the occurrence of *aerC* (a putative oxygenase catalyzing oxidation of Agm to Aeap), *aerI* (a putative glycosyltransferase catalyzing glycosylation at Choi), *aerL1* (a putative sulfotransferase catalyzing sulfation at Choi at pos. 3) and *aerL2* (putatively catalyzing sulfation at Hpla at pos. 1) and the predicted structural modification of the Aer peptide as observed among the strains.

Interestingly, *aer* gene composition, as well as its activity and the resulting Aer structural diversity, correlated with *Planktothrix* speciation processes and ecological diversification. In particular, strains of Lineage 1 were found more active in Aer production, in contrast to strains of Lineage 2, which were found frequently inactive due to several partial gene cluster deletion events or IS elements. It is concluded that various Aer peptides resulting from recombination function as chemical defense, possibly in replacement of microcystins. Thus, the Aer peptides constitute the first example of a peptide family showing a chemical diversification in relation to the ecological and phylogenetic diversification within bloom-forming cyanobacteria. In summary, the genetic variability within *aer* genes, which has been found linked to structural variability, points to adaptive significance of the Aer peptide family in the course of *Planktothrix* evolution.

## 4. Materials and Methods

### 4.1. Organisms

In this study, 124 clonal *Planktothrix* strains were analyzed. The strains were clonally isolated, characterized and assigned phylogenetically, as described previously [6,24]. They were grown under sterile conditions in BG11 medium [34] at low light intensity (5–10 µmol m^−2^s^−1^ Osram Type L30W/77 Fluora, 16/8 h light-dark cycle) at 15 °C (Lineages 1 and 2) or 23 °C (Lineage 3), Appendix A. Taxonomic affiliation was performed according to Suda et al. [35], Kurmayer et al. [24], Gaget et al. [36].

### 4.2. DNA Extraction, PCR and Sequencing

Cells from cultures were harvested by centrifugation and washed in Tris-EDTA (TE) buffer. Genomic DNA was extracted using the CTAB protocol as described previously [23]. DNA extracts were stored at −20 °C. PCR (Appendix A, primer pairs a–x) was used to determine *aer* gene presence/absence via 2 kbp amplicons, covering the entire *aer* BGC amplifying overlapping fragments without interruption. Primers were designed according to the reference *aer* gene sequences from *Planktothrix* NIVA-CYA126/8 (NZ_ASAK00000000.1) and No66 (NZ_LR882963.1). PCR mixtures had a volume of 10 µL, containing 2 μL (5×) Phusion HF Buffer (Thermo Scientific, Vienna, Austria), 500 nM of each primer, 200 μM of each deoxynucleotide triphosphate (Thermo Scientific), 0.1 U of Phusion High-Fidelity DNA Polymerase (Thermo Scientific) and 10 ng of genomic DNA as a template. PCR amplification was performed by initial denaturation at 98 °C for 1 min; thereafter, 35 cycles of denaturation at 98 °C for 10 s, annealing at variable temperature for 15 s and elongation at 72 °C (Appendix A) were performed. The PCR amplicon size was determined using standard gel electrophoresis (0.8% agarose gels in 0.5 × Tris-borate-EDTA (TBE) buffer) and visualized using Midori Green.

From the *Planktothrix* strains No63, No253, No790, No980 and No1020, *aer* gene fragments were sequenced from PCR products using standard Sanger sequencing at Eurofins Genomics (Ebersberg, Germany) via primers listed in Appendix A. For this purpose, PCR mixtures had a higher volume of 30 µL and products were extracted from the agarose gel using a commercial gel extraction kit (QIAquick, Gel Extraction Kit, Qiagen, Hilden, Germany). All new sequences obtained during this study have been included in Appendix A.

### 4.3. Sequence Comparison and Phylogenetic Analysis

The BLASTn algorithm was used to identify *aer* genes or homologs in *Planktothrix* and other cyanobacterial genera. Nucleotide sequence analysis was performed in Mega 7.0 [37]. Sequences were aligned using Clustal W (codons). To estimate the nucleotide similarity of the genes/ORFs (Figure 1), distance values were calculated using default adjustments (variance estimation: bootstrap method, 1000 replications; substitution model: nucleotide, Tamura-Nei model, transitions + transversions; uniform rates; pairwise deletions). Maximum likelihood trees (Appendix A) were calculated in Mega 7.0 (phylogeny test: bootstrap method; 100 replications; substitution model: Tamura-Nei model; uniform rates; gaps included).

For *Planktothrix*, the following *aer* sequences were obtained from NCBI, including the strains CCAP1459/11A (NZ_BJCD01000048.1), NIES-204 (AP017991.1), NIVA-CYA15 (NZ_KE734694.1), NIVA-CYA34 (NZ_AVFT01000027.1), NIVA-CYA56/3 (NZ_KE734731.1), NIVA-CYA98 (AM990465.1), NIVA-CYA126/8 (LR882934.1), NIVA-CYA405 (NZ_KE734708.1), NIVA-CYA406 (NZ_KE734710.1), NIVA-CYA407 (NZ_KE734717.1), NIVA-CYA540 (NZ_KE734720.1), No2A (LR882938.1), No66 (LR882963.1), No82 (OW445656.1), No108 (LR882941.1), No365 (LR882944.1), No758 (LR882949.1), No976 (LR882952.1), PCC7805 (LR882950.1), PCC7811 (LR882969.1), PCC7821 (LR882958.1) and PCC11201 (NZ_LT797710.1). For *M. aeruginosa*, the *aer* sequences were obtained from strain FACHB-1757 (CP011339), FD4 (CP046973.1), NIES-98 (FJ609416.1), NIES-102 (AP019314.1), NIES-298 (CP046058.1), NIES-843 (AP009552.1), NIES-2481 (NZ_CP012375.1), NIES-2549 (CP011304.1) and PCC7806 (CP130696.1). In addition, *Nodularia* strain CCY9414 (NZ_CP007203.1) and *Nostoc* strain UIC10630 (NZ_JAAGOG010000094.1) were used as outgroups.

### 4.4. Multivariate Statistical Analysis

Principal Component Analysis (PCA) was performed to cluster all 124 *Planktothrix* strains according to *aer* BGC type, as inferred from PCR results on *aer* gene presence/absence (a binary (1/0) matrix was analyzed using the statistical package IBM SPSS Statistics 24.0) (Appendix A).

### 4.5. Aeruginosin Peptide Identification and Fragmentation Using HPLC-MS^n^

Cyanobacterial cells were harvested by filtration on glass fiber filters (BMC, Ederol, Vienna, Austria), freeze-dried and stored at −20 °C. Peptides were extracted from biomass (2–5 mg dry weight) on ice with 50% aqueous methanol (*v/v*) according to [38]. Peptides were separated on a LiChroCART 250-4 cartridge system (LiChrospher 100 C18 (column dimensions 250 × 4 mm; 5 µm particle size; Merck, Darmstadt, Germany) by HPLC (HP 1100, Agilent) using a linear water/acetonitrile (0.05% trifluoroacetic acid) gradient from 20 to 50% acetonitrile in 45 min (1 mL min^−1^ flow rate, 30 °C column oven temperature). Peptide masses were determined using an ESI-MS ion trap (amazon SL, Bruker), operated in positive ion mode, with nitrogen as sheath gas (43 psi, 8 L min^−1^, 300 °C), helium as auxiliary gas and a capillary voltage of 5 kV.

In the first step, ESI-MS was coupled to HPLC for full mass scan (50–2000 *m*/*z*) and molecule automated fragmentation from the two most abundant ions and MS3 fragmentation from the most abundant ion of MS2 fragmentation spectra. Putative Aer peptides were identified by their fragmentation pattern, notably [M+H]^+^ 140 or 122, indicating the Choi-immonium ion. All putative Aer that contributed ≥5% of peak area compared to the most abundant peptide in the base peak chromatogram were recorded (Appendix A). Some Aer variants had equal masses but different retention time, often co-occurring in one strain: collection and re-injection confirmed retention time for individual Aer variants with the same protonated mass and fragmentation pattern (e.g., Appendix A), indicating structural isomers. In a second step, putative Aer fractions were purified and directly injected into ESI-MS^n^ for specific manual fragmentation (Appendix A). For example, the Aer peptide of No1020 was identified via comparison to *M. aeruginosa* NIES-89: retention time 6.9 min, [M+H]^+^ 717.3, identical fragmentation pattern and the occurrence of a triple-peak due to tautomerization (Appendix A).

### 4.6. Aeruginosin Peptide Purification and Structure Identification Using HRMS

Six unknown chlorinated Aer variants were purified (approx. 100 µg) from biomass of *Planktothrix* strains No66, NIVA-CYA116, No1020. For this purpose, strains were cultivated at the mass culture facility at CyanoBiotech GmbH (Berlin, Germany). Biomass (approx. 3 g each) was extracted on ice in an Erlenmeyer flask (250 mL) with 50% aqueous methanol (*v*/*v*) for one hour. The crude extract was centrifuged (3700 g for 15 min; Thermo Scientific Multifuge X3R) and subsequently concentrated through solid phase extraction (SPE) via C18 cartridges (Sep-Pak ^®^ tC18 cartridges; Waters, Vienna, Austria). The supernatant was filtered after centrifugation using glass microfiber filters (GF/F filters; 0.7 μm pore size; Whatman; 47 mm diameter). The filtrate was diluted with MilliQ water to 10% aqueous methanol and transferred to a phase-separation funnel. The SPE columns were activated with 100% methanol (3 × 1 mL) and equilibrated with 10% methanol (3 × 1 mL). The diluted crude extracts were applied after conditioning of the SPE columns under low pressure with a flow-through rate of 1 mL min^−1^ and eluted with 1 mL of 80% aqueous methanol (*v*/*v*).

Purified Aer were analyzed using an Ultra-High Performance Liquid Chromatography coupled to a Xevo quadrupole time-of-flight mass spectrometer (Waters, Wilmslow, UK). Peptides were separated on a BEH C18 column (100 × 2.1 mm; 1.7 μm particle size) which was maintained at 40 °C. The mobile phase was mixed by Milli-Q water plus 0.1% formic acid and acetonitrile plus 0.1% formic acid, with a linear gradient increasing from 20% to 70% acetonitrile within 10 min. Data were acquired by positive ion electrospray scanning from 50 to 2000 *m*/*z* with a scan time of 2 s and inter-scan delay of 0.1s. Ion source parameters, i.e., capillary and sampling cone, were 2.9 V and 25 V, respectively; desolvation temperature, 300 °C; and source temperature, 80 °C. Cone and desolvation gas flows were 50 L h^−1^ and 400 L h^−1^ respectively. Sodium iodide (2 μg μL^−1^ in 50/50 Propan-2-ol/Milli-Q) was used as calibrant with Leucine-enkephalin (0.5 mg mL^−1^ in 50/50 methanol/Milli-Q) as reference mass. Instrument control and processing were achieved using MassLynx v4.1. MS/MS spectra for the individual Aer molecule were obtained by targeting the protonated parent at 20–50 eV.

### 4.7. Toxicity Tests

To test the toxicity of the new peptides Aer 716 and Aer 688, several toxic/bioactive peptides produced by *Planktothrix*, i.e., [D-Asp^3^, (E)-Dhb^7^] MC-RR and anabaenopeptin B and F, as well as Aer 828A [30] were purified and tested using the Thamnotoxkit F (MicroBio Tests, Belgium). This test is a 24 h crustacean toxicity bioassay using larvae of the crustacean *T. platyurus*. The test was performed following the manufacturer’s instructions and a standardized operational procedure [39].

The concentration of the purified compounds was quantified via the molar extinction coefficient by measuring the absorbance using a spectrophotometer (UVmini-1240, Shimadzu). For the Aer peptides, the concentration was calculated as aeruginosin 103A equivalents (λ_max_ = 224 nm; ε = 11,600) [40]. Anabaenopeptins B and F were quantified as anabaenopeptin B equivalents (λ_max_ = 225 nm; ε = 8833) [41]. [D-Asp^3^, (E)-Dhb^7^] MC-RR was quantified using the extinction coefficient 50,400 at 239 nm (λ_max_) [42]. The compounds were dissolved in 1 mL 100% methanol. Each compound was tested in several concentrations in triplicates at least, with 10–20 animals per approach, i.e., [D-Asp^3^, (E)-Dhb^7^] MC-RR (1–10 µM), Aer 828A, Aer 716, Aer 688 (5–100 µM), anabaenopeptin B and F (5–120 µM). The final concentration of methanol did not exceed 1%. The mortality in the standard freshwater blanks and 1% methanol blanks was not allowed to exceed 10%.

## Figures and Tables

**Figure 1 marinedrugs-21-00638-f001:**
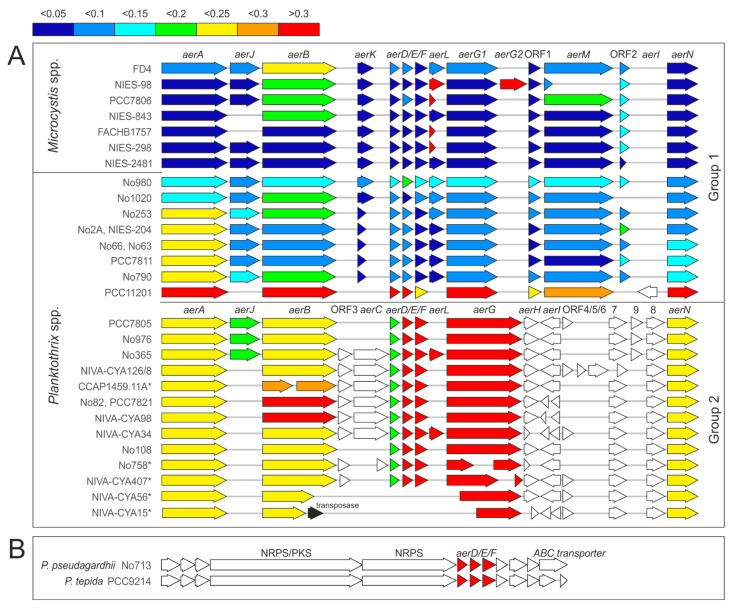
(**A**) Genetic basis for aeruginosin synthesis as compared between *Microcystis* and *Planktothrix*. The colors illustrate the genetic distance to *Microcystis* strain NIES-2549: lowest distance (dark blue) to high distance (in red). *, strains carrying *aer* genes but not producing aeruginosin. Additional sequences for *Microcystis* and *Planktothrix* strain PCC11201 have been included [27]). (**B**) Putative NRPS/PKS genes possibly encoding another Choi moiety similar to the aeruginosin peptide family.

**Figure 2 marinedrugs-21-00638-f002:**
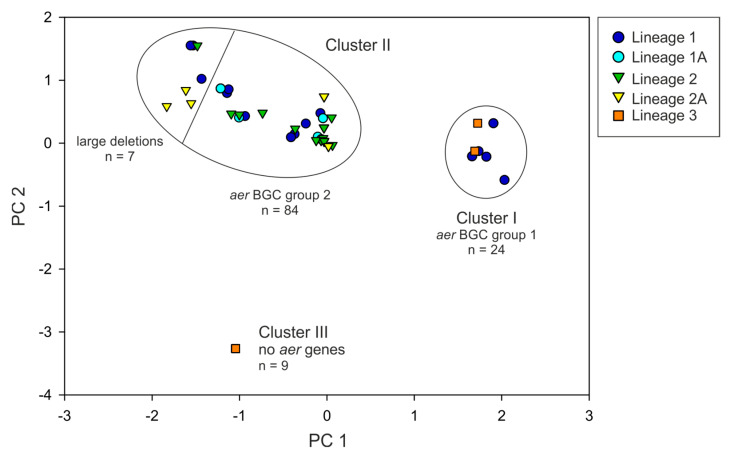
Ordination of 124 *Planktothrix* strains according to *aer* gene presence/absence via principal component analysis (PCA). The *aer* gene (fragment) presence or absence was determined via PCR, amplifying 2 kbp-sized fragments of the *aer* gene cluster without interruption. Positive or negative PCR results were indicated as a binary (1/0) matrix.

**Figure 3 marinedrugs-21-00638-f003:**
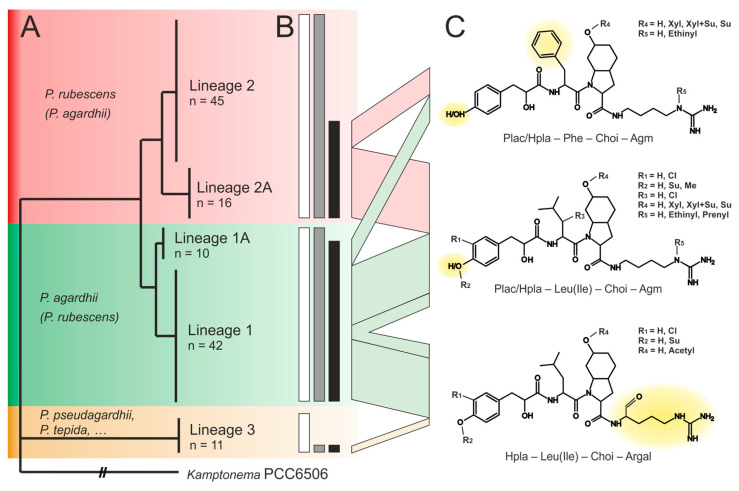
(**A**) Phylogenetic distribution of aeruginosin-producing strains in *Planktothrix* spp. (*n* = 124). The number of strains assigned to Lineages 1, 2, 3 is indicated as described previously [23]. (**B**) The bar charts indicate the proportion of strains carrying *aer* genes (in grey) relative to analyzed strains (in white) and found active in Aer synthesis (in black). The width of the arrows is proportional to the number of strains carrying a specific aeruginosin variant. (**C**) Aeruginosin tetrapeptide structures and accessory modifications (yellow marked moieties highlight the structural differences in the backbone).

**Figure 4 marinedrugs-21-00638-f004:**
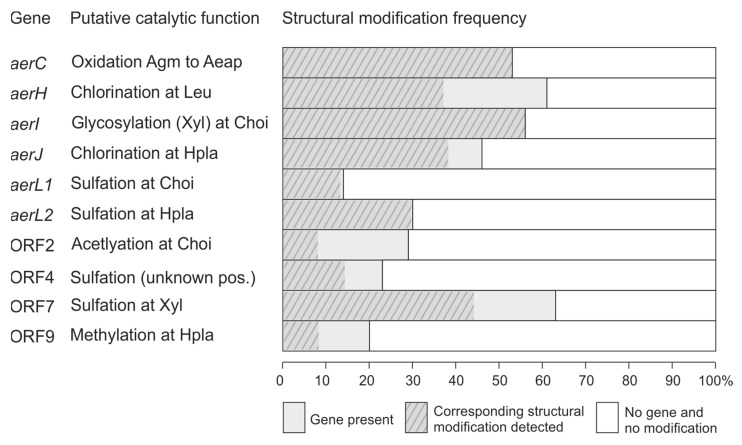
Frequency of occurrence of accessory gene functions and corresponding catalyzed structural modifications in aeruginosins as observed among 79 *Planktothrix* spp. strains. There was a perfect correlation between *aerC*, *aerI*, *aerL1*, *aerL2* gene presence and the predicted structural modification.

**Table 1 marinedrugs-21-00638-t001:** Functions of *aer* genes as published or indicated during this study. Abbreviations: NRPS, non-ribosomal peptide synthetase; PKS, polyketide synthase; Agm, agmatine; Aeap, 1-amino-2-(N-amidino-D3-pyrrolinyl)-ethyl; Plac, phenyllactic acid; Hpla, hydroxyphenyllactic acid; Xyl, xylose.

Gene	Functional Description	Literature
*aerA*	NRPS/PKS hydrid (incorporation of Plac/Hpla at pos. 1)	[13,21]
*aerB*	NRPS (incorporation of Leu/Ile/Phe at pos. 2)	[13,21]
*aerC*	oxygenase (oxidation of Agm to Aeap)	[14]; this study ^1^
*aerD*, *E*, *F*	Choi synthesis (isomerase, reductase)	[11,12,13]
*aerG*, *G1* + *G2*, *G1* + *M*	NRPS (Choi at pos. 3 and Agm derivate at pos. 4)	[13,21]
*aerH*	halogenase (chlorination at Leu at pos. 2)	this study
*aerI*	glycosyltransferase (Xyl at Choi)	[13,21]
*aerJ*	halogenase (chlorination at Hpla at pos. 1)	[21,28,29]; this study
*aerK*	isomerase	[14]
*aerL1*	sulfotransferase (sulfation at Choi at pos. 3)	[21]; this study
*aerL2*	sulfotransferase (sulfation at Hpla at pos. 1)	[21]; this study
*aerN*	ABC transporter	[13,21]
ORF1	oxidoreductase (reduction of Agm; cleavage from synthesis operon)	[14]; this study
ORF2	hypothetical protein (*O*-acetylation at Choi)	this study
ORF3	hypothetical protein	this study
ORF4	sulfotransferase (sulfation at unknown pos.)	[14]; this study
ORF5	hypothetical protein	
ORF6	hypothetical protein	
ORF7	sulfotransferase (sulfation at Xyl)	[14]; this study
ORF8	aldo-/keto reductase (reduction of Agm; cleavage from synthesis operon)	[14]; this study
ORF9	hypothetical protein (*O*-methylation at Hpla at pos. 1)	this study

^1^ putative function of accessory *aer* genes inferred from correlation between gene presence and corresponding structural modification (see main text).

**Table 2 marinedrugs-21-00638-t002:** Overview of purified chlorinated aeruginosins, resulting from the large-range recombination event (*aer* BGC group 1), and the respective HRMS results. The presence of the Choi fragment and chlorine is indicated. The proposed sum formula for the chlorinated aeruginosin is indicated (nd, not detected).

Strain	Retention Time [min]	[M+H]^+^	Choi Fragment [M+H]^+^	Chlorine Detected	Sum Formula
NIVA-CYA116	4.1	717.25	140.1	Yes	nd
	5.9	689.25	140.1	Yes	nd
	12.5	759.26	122.1	Yes	nd
	16.5	731.28	122.1	Yes	nd
No1020	6.6	717.25	140.1	Yes	nd
No66	6.3	717.20	140.1	Yes	C_30_H_46_N_6_O_10_SCl

## Data Availability

The original data that support the findings of this study are available from the corresponding authors upon reasonable request.

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
