# Peer review of "High Structural Diversity of Aeruginosins in Bloom-Forming Cyanobacteria of the Genus Planktothrix as a Consequence of Multiple Recombination Events"

_marinedrugs, 2023, doi:10.3390/md21120638_

Round 1
Reviewer 1 Report
Comments and Suggestions for Authors
The authors focus on high structural diversity of aeruginosins in bloom-forming cyanobacteria of the genus Planktothrix. The manuscript has potential but is not currently publishable until the comments below are adequately addressed. My decision is major revise.
Abstract
1. The abstract should clearly indicate the relevance of the work for international research, the main findings and the contribution of this study. Please authors rewrite this part.
Keywords
1. Please authors think carefully about keywords.
Can “horizontal gene transfer (HGT)” be changed as “horizontal gene transfer”?
What is the full name of cyanoHABs? It did not appear in the manuscript.
Introduction
1. Line 46-47: “Genome sequencing and metabolomics have shown that the Aer peptide family is more widely distributed than previously assumed.” Please add references related to genome sequencing and metabolomics.
2. Some studies on cyanobacteria can be added in this part.
Chen, L., Giesy, J.P., Adamovsky, O., Svirčev, Z., Meriluoto, J., Codd, G.A., Mijovic, B., Shi, T., Tuo, X., Li, S.C., Pan, B.Z., Chen, J., Xie, P., 2021. Challenges of using blooms of Microcystis spp. in animal feeds: a comprehensive review of nutritional, toxicological and microbial health evaluation. Sci Total Environ 764, 142319.
Results and Conclusions
1. Line 99: What is the full name of ORF2?
2. Line 149: “115 strains (91%)” Is the total 124 strains?
3. Line 187: “We found 79 Planktothrix strains (64% of all strains) as Aer producers.” Please list the figure or table of this result? (The basis for this result)
4. The authors need to add some related researches and discussions in order to indicate the relevance of the work for international.
References
1. The number of the references is relatively small and there are not many recent studies.
Table 1
1. In this manuscript, how did the authors identify the function of aer genes such as aerH, ORF2, ORF3, ORF5, ORF6 and ORF9?
Comments on the Quality of English Language
Moderate editing of English language required.
Author Response
Reviewer 1
The authors focus on high structural diversity of aeruginosins in bloom-forming cyanobacteria of the genus Planktothrix. The manuscript has potential but is not currently publishable until the comments below are adequately addressed. My decision is major revise.
Abstract
- The abstract should clearly indicate the relevance of the work for international research, the main findings and the contribution of this study. Please authors rewrite this part.
Response: We rewrote the abstract in the hope that we implemented the suggestions well.
Keywords
- Please authors think carefully about keywords.
Can “horizontal gene transfer (HGT)” be changed as “horizontal gene transfer”?
What is the full name of cyanoHABs? It did not appear in the
Response: We revised the keywords according to the suggestions.
Introduction
- Line 46-47: “Genome sequencing and metabolomics have shown that the Aer peptide family is more widely distributed than previously assumed.” Please add references related to genome sequencing and metabolomics.
Response: We added relevant references to support the information obtained through genome sequencing and metabolomics.
- Calteau, A.; Fewer, D. P.; Latifi, A.; Coursin, T.; Laurent, T.; Jokela, J.; Kerfeld, C. A.; Sivonen, K.; Piel, J.; Gugger, M., Phylum-wide comparative genomics unravel the diversity of secondary metabolism in Cyanobacteria. BMC Genomics 2014, 15, (1), 977. doi: 10.1186/1471-2164-15-977.
- Ahmed, M. N.; Wahlsten, M.; Jokela, J.; Nees, M.; Stenman, U. H.; Alvarenga, D. O.; Strandin, T.; Sivonen, K.; Poso, A.; Permi, P.; Metsä-Ketelä, M.; Koistinen, H.; Fewer, D. P., Potent Inhibitor of Human Trypsins from the Aeruginosin Family of Natural Products. ACS Chem Biol 2021, 16, (11), 2537-2546. doi: 10.1021/acschembio.1c00611.
- Some studies on cyanobacteria can be added in this part.
Chen, L., Giesy, J.P., Adamovsky, O., Svirčev, Z., Meriluoto, J., Codd, G.A., Mijovic, B., Shi, T., Tuo, X., Li, S.C., Pan, B.Z., Chen, J., Xie, P., 2021. Challenges of using blooms of Microcystis spp. in animal feeds: a comprehensive review of nutritional, toxicological and microbial health evaluation. Sci Total Environ 764, 142319.
Response: We did not include this reference because it does not appear to be relevant.
Results and Conclusions
- Line 99: What is the full name of ORF2?
Response: We wrote the full name of ORF2 (open reading frame 2).
- Line 149: “115 strains (91%)” Is the total 124 strains?
Response: Thank you for pointing out the incorrect calculation. We have corrected this.
“Out of all 124 strains the aer genes were detected for 115 strains (93%).”
- Line 187: “We found 79 Planktothrix strains (64% of all strains) as Aer producers.” Please list the figure or table of this result? (The basis for this result)
Response: We added Table S3 as source for this result.
“We found 79 Planktothrix strains (64% of all strains) as Aer producers (Table S3).”
- The authors need to add some related researches and discussions in order to indicate the relevance of the work for international.
Response: Information has been added under section 2.6 (toxicity of aeruginosins) to show its relevance.
References
- The number of the references is relatively small and there are not many recent studies.
Response: We added relevant references (41 vs 32 references).
Table 1
- In this manuscript, how did the authors identify the function of aer genes such as aerH, ORF2, ORF3, ORF5, ORF6 and ORF9?
Response: We added a footnote to Table 1.
“1 putative function of accessory aer genes inferred from correlation between gene presence and corresponding structural modification (see main text and Figure 4).”
Reviewer 2 Report
Comments and Suggestions for Authors
This research was done at a high scientific level. The findings from the study are fully and appropriately displayed. Very interesting conclusions were reached, including the finding that genetic heterogeneity within aer genes, which has been connected to structural variability, indicates the adaptive significance of the Aer peptide family during Planktothrix evolution.
There have been no serious comments on the article. It should only be noted that:
1) The aim of the study should be stated clearly and succinctly at the end of the Introduction section;
2) There is a lot of information in the "Conclusion" section that is more appropriate for the "Discussion of Results" section. The “Conclusion” section should present the main conclusions that the authors came to in the study, as well as describe the possible consequences arising from the study and outline possible prospects for further research on this topic. The "Conclusions" section should no longer contain any literary references; instead, the related ideas should be moved to the "Discussion" section.
3) In the section “Materials and Methods” (line 370) a reference should be made to a literary source that describes the medium BG-11 for the cultivation of cyanobacteria.
Author Response
Reviewer 2
This research was done at a high scientific level. The findings from the study are fully and appropriately displayed. Very interesting conclusions were reached, including the finding that genetic heterogeneity within aer genes, which has been connected to structural variability, indicates the adaptive significance of the Aer peptide family during Planktothrix evolution.
There have been no serious comments on the article. It should only be noted that:
1) The aim of the study should be stated clearly and succinctly at the end of the Introduction section;
Response: We added a sentence to the aim of this study in the introduction section:
“It was the aim of this study to describe all the recombination events resulting in Aer structural modification in relation to Planktothrix spp. eco-evolutionary divergence.”
2) There is a lot of information in the "Conclusion" section that is more appropriate for the "Discussion of Results" section. The “Conclusion” section should present the main conclusions that the authors came to in the study, as well as describe the possible consequences arising from the study and outline possible prospects for further research on this topic. The "Conclusions" section should no longer contain any literary references; instead, the related ideas should be moved to the "Discussion" section.
Response: Thank you for this suggestion. We transferred sentences to the results and discussion section to shorten the conclusion section.
3) In the section “Materials and Methods” (line 370) a reference should be made to a literary source that describes the medium BG-11 for the cultivation of cyanobacteria.
Response: We added the reference for BG11 medium.
“Rippka, R., Isolation and purification of cyanobacteria. Methods Enzymol 1988, 167, 3-27. doi: 10.1016/0076-6879(88)67004-2.”
Round 2
Reviewer 1 Report
Comments and Suggestions for Authors
The revised manuscript was received but there are some questions as follows. Before it can be accepted for publication a revision is needed.
Introduction
1. Line 48-51: “In contrast to other NRPS pathways (i.e. microcystins, anabaenopeptins) numerous facultative tailoring enzymes (e.g., sulfatases, halogenases, glycosyltransferases, methyltransferases, acetyltransferases) introduce high structural variability into the Aer peptide family”
Please add the related reference. What is the NRPS pathway?
Results and Conclusions
1. Line 141-l43: “Abbreviations: NRPS, non-ribosomal peptide synthethase, PKS, polyketide synthase, Agm = agmatine, Aeap = 1-amino-2- (N-amidino-D3-pyrrolinyl)-ethyl, Plac = phenyllactic acid, Hpla = hydroxyphenyllactic acid, Xyl = xylose.” could change as “Abbreviations: NRPS, non-ribosomal peptide synthethase; PKS, polyketide synthase; Agm, agmatine; Aeap, 1-amino-2- (N-amidino-D3-pyrrolinyl)-ethyl; Plac, phenyllactic acid; Hpla, hydroxyphenyllactic acid; Xyl, xylose.” ?
2. Line 237: What is the full name of “AA”?
3. Line 266: “by the same authors a sulfate group was found linked to the xylose moiety.” Please add the related reference.
4. Line 333-335: “However, by the same authors, conformational changes resulting from a chlorine towards both increasing or decreasing the bioactivity/toxicity also have been suggested.” Please add the related reference.
5. Line 358: What is the full name of “IS”?
Materials and Methods
1. Line 468: “0.1% formic acid (B)”
Line 480: “anabaenopeptin B”
Table 1 and table 2
1. The format of table 1 and table 2 is incorrect. A three-line table is recommended, such as table 1 in previous manuscript.
2. In table 1 and figure 3: Is “Ile” consistent with “Ile” in the manuscript (such as line 102)?
3. Table 2: Line 310-311: “The presence of the Choi fragment and Cl- is indicated.” But it is “Cl detected” in Table 2.
Figure
1. Figure 1 and figure 3: Is “aerI” in consistent with “aerI” in the manuscript (such as Line 105)
Comments on the Quality of English Language
Moderate editing of English language required
Author Response
Dear editors,
Please find enclosed our revised MS accordingly to the suggestions of reviewer 1.
Most importantly:
- The most recent version has been revised using track and trace changes and the comments of reviewer 1 have been followed as indicated.
- All references have been re-checked to include proper citation.
As our co-author Prof. Christine Edwards has checked the English language, we did not invoke a further (commercial) proof reading service.
In addition in correspondence it has been suggested by you that your in-house English editor will help double-check and proof the English language.
Yours sincerely
Rainer Kurmayer
The revised manuscript was received but there are some questions as follows. Before it can be accepted for publication a revision is needed.
Introduction
- Line 48-51: “In contrast to other NRPS pathways (i.e. microcystins, anabaenopeptins) numerous facultative tailoring enzymes (e.g., sulfatases, halogenases, glycosyltransferases, methyltransferases, acetyltransferases) introduce high structural variability into the Aer peptide family”
Please add the related reference. What is the NRPS pathway?
Reference [14] has been added, it is now stated: “In contrast to other peptides (i.e. microcystins, anabaenopeptins), numerous tailoring enzymes…..[14].”
Results and Conclusions
- Line 141-l43: “Abbreviations: NRPS, non-ribosomal peptide synthethase, PKS, polyketide synthase, Agm = agmatine, Aeap = 1-amino-2- (N-amidino-D3-pyrrolinyl)-ethyl, Plac = phenyllactic acid, Hpla = hydroxyphenyllactic acid, Xyl = xylose.” could change as “Abbreviations: NRPS, non-ribosomal peptide synthethase; PKS, polyketide synthase; Agm, agmatine; Aeap, 1-amino-2- (N-amidino-D3-pyrrolinyl)-ethyl; Plac, phenyllactic acid; Hpla, hydroxyphenyllactic acid; Xyl, xylose.” ?
We changed this. - Line 237: What is the full name of “AA”?
We changed AA to “amino acid”. - Line 266: “by the same authors a sulfate group was found linked to the xylose moiety.” Please add the related reference.
We added the reference. - Line 333-335: “However, by the same authors, conformational changes resulting from a chlorine towards both increasing or decreasing the bioactivity/toxicity also have been suggested.” Please add the related reference.
We added the reference. - Line 358: What is the full name of “IS”?
We added “insertion sequence” as full name of IS.
Materials and Methods
- Line 468: “0.1% formic acid (B)”
We revised the sentence: “The mobile phase was mixed by Milli-Q water plus 0.1% formic acid and acetonitrile plus 0.1% formic acid, with a linear gradient increasing from 20% to 70% acetonitrile within 10 min.”
Line 480: “anabaenopeptin B”
This is the name of the peptide anabaenopeptin B.
Table 1 and table 2
- The format of table 1 and table 2 is incorrect. A three-line table is recommended, such as table 1 in previous manuscript.
We adjusted the format of Table 1 and 2. - In table 1 and figure 3: Is “Ile” consistent with “Ile” in the manuscript (such as line 102)?
Yes it is. “Ile” stands for isoleucine. - Table 2: Line 310-311: “The presence of the Choi fragment and Cl- is indicated.” But it is “Cl detected” in Table 2.
Yes, we detected the chlorine (Cl) at the peptides according to the characteristic isotope patterns. The choi fragment was either detected as 140 or 122 m/z, depending on the additional group at the hydroxygroup of the choi, that is preferred cut off with or without OH2.
Figure
- Figure 1 and figure 3: Is “aerI” in consistent with “aerI” in the manuscript (such as Line 105)
Yes, “aerI” in figure 1 and figure 4 is consistent with “aerI” in the text.
End of response